# Presumed Bilateral Ciliary Body Medulloepithelioma in a Child with Pleuropulmonary Blastoma and *DICER1* Mutation

**DOI:** 10.3390/diagnostics15060694

**Published:** 2025-03-11

**Authors:** Małgorzata Danowska, Anna Rogowska, Krzysztof Cieślik, Joanna Jędrzejczak-Młodziejewska, Klaudia Rakusiewicz-Krasnodębska, Wojciech Hautz

**Affiliations:** Ophthalmology Department, Children’s Health Memorial Institute, 04-736 Warsaw, Poland; a.rogowska@ipczd.pl (A.R.); k.cieslik@ipczd.pl (K.C.); j.jedrzejczak-mlodziejewska@ipczd.pl (J.J.-M.); k.rakusiewicz@ipczd.pl (K.R.-K.); w.hautz@ipczd.pl (W.H.)

**Keywords:** ciliary body medulloepithelioma, *DICER1*, pleuropulmonary blastoma, fundus photography, tumor predisposition syndrome

## Abstract

**Background and clinical significance**: Ciliary body medulloepithelioma (CBME) is a rare germinal tumor deriving from nonpigmented ciliary epithelium, usually occurring during the first decade of life. Typically, the diagnosis is delayed as a result of the tumor’s slow growth and late onset of symptoms. Primary enucleation is commonly required; nevertheless, globe-sparing means of therapy have been successful in selected cases. CBME is among the spectrum of neoplasms associated with *DICER1* cancer predisposition syndrome. **Case presentation**: Herein, we report a case of a 6-year-old boy with a history of pleuropulmonary blastoma type II and *DICER1* mutation who presented with leukocoria in his right eye. After a thorough ophthalmological evaluation, he was diagnosed with CBME. Because of the large size of the lesion and vitreous seeding, the eye was enucleated. Histopathology confirmed the diagnosis of a benign teratoid medulloepithelioma. After 2 years of regular ophthalmological examinations, a new lesion was detected in the left eye. Three cycles of transscleral cryotherapy allowed for tumor control and globe salvage. The patient remains recurrence-free 6 months after the last treatment. **Conclusions**: This report should raise awareness among clinicians about the possibility of bilateral CBME and the necessity of regular ophthalmological screening in patients with *DICER1* syndrome, especially those previously treated for CBME.

## 1. Introduction

Ciliary body medulloepithelioma (CBME) is a rare intraocular tumor, with the majority of cases developing within the first decade of life [1,2,3,4]. It is congenital, arising from the nonpigmented ciliary epithelium or rarely from the iris, retina, or optic disc. Due to the rarity of the tumor, the exact incidence is unknown. Initially, small lesions can remain asymptomatic until they reach a size sufficient to be seen through the pupil or cause secondary ocular symptoms. The most common presentation includes loss or worsening of vision, leukocoria, redness, pain, strabismus, or visible intraocular mass [1,2,5,6].

The clinical characteristics of CBME include a whitish to light-pink intraocular mass located in the region of the ciliary body, with intratumoral cysts present in more than 50% of cases [2,4,6]. Typical local effects of the tumor include lens subluxation with a sectoral or total cataract, a bowing of the iris, corectopia, ectropion uveae, neovascularization of the iris, and the presence of the neoplastic cyclitic membrane [1,2,4,6].

The standard treatment in advanced tumors is primary enucleation [1,2,5,6]; however, treatment of smaller lesions has been attempted with globe-sparing means of therapy.

An association has been discovered between CBME and pleuropulmonary blastoma (PPB)—a rare pediatric lung malignancy most frequently occurring in individuals with *DICER1* mutation [3,7,8,9]. Patients with *DICER1* tumor predisposition syndrome are at risk of developing PPB and CBME along with ovarian sex cord-stromal tumors, brain tumors including pineoblastoma and pituitary blastoma, renal sarcoma, Wilms tumor, cystic nephroma, multinodular goiter and carcinoma of the thyroid, nasal chondromesenchymal hamartoma and embryonal rhabdomyosarcoma [9].

## 2. Case Study

A 6-year-old boy who had been treated in our Oncology Department previously presented with leukocoria in the right eye, noticed 2 days earlier.

At the age of 3 years, the patient experienced recurrent respiratory tract infections with cough and two episodes of spontaneous pneumothorax. His previous medical history had been unremarkable. Chest X-ray and computed tomography revealed emphysema in his left lung. The patient underwent a thoracoscopic lung resection. The histological diagnosis was pleuropulmonary blastoma type II, and microscopically radical resection was confirmed. After a complete diagnostic work-up, no evidence of metastasis was detected. The patient received nine courses of IVA chemotherapy (ifosfamide + vincristine + actinomycin D). No PPB recurrence was noticed during follow-up.

*DICER1* mutation had been confirmed in the boy and his mother and sister. The sister was also known to have developed thyroid hyperplasia and an iris cyst.

Upon presentation to the Ophthalmology Department at the age of 6, visual acuity was 0.1 in the right eye and 0.8 in the left (Snellen chart). The intraocular pressure taken with rebound tonometry was 10 and 12 mmHg, respectively. The anterior segment evaluation in the right eye under general anesthesia revealed mild conjunctival injection, slight anterior bowing of the iris in the supero-temporal quadrant, ectropion uveae, and a temporal posterior capsular opacification. Adherent to the posterior capsule of the lens was a pale pink mass with visible blood vessels covering over 5 clock hours (Figure 1). Fundus examination revealed flat inferior retinal detachment, a slightly elevated optic disc with bands of vitreous seeding extending from the surface of the tumor to the disc, and retinal vessel tortuosity. Ultrasonography and ultra biomicroscopy (UBM) revealed a mass of mixed echogenicity with numerous hypoechogenic spaces adjacent to the temporal part of the ciliary body with localized flat retinal detachment and hyperechogenic bands in the vitreous chamber. Examination of the left eye was unremarkable.

Magnetic Resonance Imaging (MRI) showed an irregular-surfaced ciliary body mass of heterogenic intensity, size of 13 × 11 × 14 mm (Figure 2) in the antero-lateral part of the right eye, with a heterogenous enhancement pattern. No scleral or extrabulbar invasion was revealed. Comprehensive additional examinations, including abdomen ultrasonography, chest radiograph, and blood analyses, revealed no additional abnormalities.

Due to the large dimensions of the tumor and vitreous seeding, after obtaining informed consent, immediate enucleation was undertaken. The procedure was performed under general anesthesia, with minimal manipulation of the globe. The histopathological examination disclosed a mass with cords of neuroepithelial cells with heteroplastic elements of hyaline cartilage. The lesion was diagnosed as benign teratoid medulloepithelioma.

The patient remained under careful surveillance—an ophthalmological examination under anesthesia and UBM was scheduled every 6 months. An examination 2 years post enucleation of the right eye revealed a new flat whitish mass in the infero-temporal ora serrata of the left eye (Figure 3A), without any corresponding abnormalities in the UBM. The following exam after 2 months showed an increase in the mass size with small neovascular vessels on the surface (Figure 3B) and a small hypoechogenic cyst of the ciliary body. The clinical findings supported a diagnosis of medulloepithelioma in the left eye. We proceeded with transscleral cryotherapy with a triple freeze-thaw technique every 4–6 weeks. After two cycles of cryotherapy, retinal detachment was observed due to a retinal tear at the base of the tumor. The patient was treated with scleral buckling combined with intraoperative cryotherapy. An examination a month after the procedure revealed a reattached retina and a decrease in the size of the tumor, with fibrotic changes and scarring. Visual acuity was 0.4 (Snellen chart). In further follow-up examinations at 3 and 6 months after the last cryotherapy session, the lesion remains stable and continues to be closely monitored (Figure 3C).

## 3. Discussion

To the best of our knowledge, this is the first report of presumed bilateral CBME in a patient with *DICER1* mutation and PPB. Lumbroso et al. [10] described a case of a 7-year-old boy with histopathology-proven CBME in one eye and a clinically presumed CBME treated with plaque radiotherapy in the other. Gupta et al. reported a case of bilateral CBME in a child with bilateral ectopia lentis as the only presenting factor [11]. In 2023, Hu et al. [12] described a case of histopathologically confirmed bilateral CBME in a 4.5-month-old boy treated with bilateral enucleation. None of these articles, however, included any mention of genetic testing or coexisting comorbidities.

*DICER1* is a widely expressed gene encoding ribonuclease III essential in the production of microRNAs [3,7,8,9]. Pathogenic germline *DICER1* variants were described as the genetic basis of PPB and later associated with a spectrum of benign and malignant neoplasms affecting mostly children and adolescents. These include ovarian sex-cord stromal tumors, renal tumors, pituitary blastoma, pineoblastoma, thyroid hyperplasia and carcinoma, central nervous system sarcoma, nasal chondromesenchymal hamartoma, and hamartomatous intestinal polyps. Pathogenic *DICER1* variants are found in approximately 1:2529–1:10,600 in the general population [9].

Although PPB and *DICER1* syndrome have been confirmed as risk factors for CBME, only a minority of patients with PPB will develop CBME. In a study by Priest et al. [7], CBME occurred in 3 out of 299 (1%) patients with PPB and 1 parent. In a study of 103 *DICER1* carriers, 3 (3%) developed CBME [8]. In a study by Kaliki et al. [1], 2 of 41 (5%) patients with CBME had a history of PPB. Durieux et al. [13] described one case of CBME in a patient with *DICER1* mutation without PPB. According to Huryn et al. [8], *DICER1* carriers may have an increased risk of other ocular, especially retinal abnormalities.

A diagnosis of CBME is often delayed because the tumor usually remains asymptomatic until its advanced size produces secondary symptoms. In a study by Kaliki et al. [1], cataracts were present in 46% of cases and glaucoma in 44%. Eighty-eight percent of cases were initially misdiagnosed as other conditions. Broughton and Zimmermann [2] also reported cataracts and secondary glaucoma as the most frequently associated ocular findings in 17 and 18 out of 56 patients, respectively. In a cohort of 16 CBME patients described by Canning et al. [5], the most frequent presenting signs were rubeosis (13 patients), cataracts (8), and glaucoma (8). In the group of 11 patients reported by Yi et al. [6], 3 were initially misdiagnosed and received treatment for glaucoma or cataracts. A total of 3/4 cases reported by Priest et al. [7] were associated with cataracts. Similarly, upon diagnosis, our patient’s right eye manifested partial posterior capsular opacification, however, without any signs of glaucoma.

The typical clinical features of CBME include a ciliary body mass with intratumoral cysts present in over 50% of cases [2,4,6] and also observed in our patient’s right eye during UBM examination. Retinal detachment noted in our patient was previously described in a minority of cases—in 3/56 patients by Broughton and Zimmermann [2], 2/16 patients by Canning et al. [5], and 3/11 cases by Yi et al. It has been, however, reported in patients with CBME and *DICER1* mutation [8,13]. A neoplastic cyclitic membrane, commonly described in other reports [1,2,4,13], was not noted in our patient.

Histopathologically, the tumor can be classified as malignant or benign. The most commonly used criteria for malignancy derived by Broughton and Zimmermann [2] and later used by Shields [4] and Kaliki [1] include areas of poorly differentiated neuroblastic cells, high pleomorphism or mitotic activity, sarcomatous areas, and invasion of the uvea, cornea, or sclera with or without extraocular extension. However, most malignant tumors do not have a tendency to metastasize unless extraocular extensions occur [1,2,4]. Most likely because of the rarity of the disease and enucleation being the most frequent treatment of choice, we have not come across any reports of a malignant transformation of a primarily benign CBME. Nineteen cases of benign CBME described by Broughton and Zimmermann [2] were small tumors confined to the inner surface of the ciliary body, the posterior chamber, or the retrolental area. Interestingly, despite being classified as histologically benign, the tumor in our patient’s right eye was extensive and manifested vitreous seeding. Regardless of the malignancy status, CBME can be classified as teratoid or non-teratoid based on the presence of heteroplastic elements in the proliferation of primitive medullary epithelium. The most frequently observed heteroplastic element in previous reports was hyaline cartilage [1,2], as was found in our patient.

Since *DICER1* syndrome predisposes affected individuals to multiple neoplasms throughout their lifetime, the International *DICER1* Symposium recommends surveillance strategies for *DICER1*-associated pulmonary, renal, gynecologic, thyroid, ophthalmologic, otolaryngologic, and central nervous system tumors and gastrointestinal polyps. A dilated ophthalmological exam (generally unsedated) is recommended annually for all patients with *DICER1* mutation aged 3–10 years [9]. However, because of the rarity of the tumor, no clear surveillance guidelines for patients with a history of CBME in one eye are provided.

CBME has a relatively slow growth pattern [2,4]. In a report by Huryn et al. [8], patients with known *DICER1* mutation received a thorough ophthalmic examination. Two children developed CBME 4,5 and 5 years afterward. Gupta et al. [11] report a case with an interval of 5 years between initial symptoms and the diagnosis of CBME. UBM has been proven to be a valuable tool in the detection and monitoring of small ciliary body tumors <4 mm [14]. Considering the patient’s age and expected difficulty with cooperation, we implemented a regimen of ophthalmological examinations with UBM under general anesthesia every 6 months.

A literature review has revealed no reports of patients undergoing ophthalmological screening focused on CBME, making our case unique. Detection of a small tumor, however, introduces the challenge of avoiding enucleation, which in most reports is described as a standard method of treatment. In the two largest published CBME case series by Kaliki et al. [1] and by Broughton and Zimmermann [2], enucleation was performed in 21/41 cases and 51/56 cases, respectively. Also, in the two smaller cohorts described by Canning et al. [5] and Yi et al. [6], 16/16 and 10/11 eyes were eventually enucleated, and the authors recommend this method of therapy for all tumors but the most circumscribed. In a study by Priest et al. [7], two out of three patients with CBME and PPB underwent enucleation.

Several globe-sparing therapeutic approaches have been described in individual cases or small groups of patients, with varying success rates.

Ang et al. [15] described a series of six cases treated with plaque brachytherapy for CBME, which allowed for globe salvage in four out of six patients with no metastases or deaths during a mean follow-up of 59 months. However, this treatment modality did not achieve sufficient tumor control as a secondary procedure in a study by Shields et al. [4].

Local resection has shown a limited success rate. In a study by Kaliki et al., local partial lamellar sclerouvectomy was associated with 50% of tumor recurrence [1], while Shields et al. reported a 100% rate of CBME recurrence after local resection [4]. Broughton and Zimmermann reported secondary enucleation in 80% of patients after initial iridectomy or iridocyclectomy [2]. According to Canning et al. [5], all four patients who underwent initial iridocyclectomy eventually required enucleation.

A single case of globe salvage in CBME treated with combined intracameral and intravitreal melphalan injections has been reported [16].

In a case series by Shields et al. [4], cryotherapy was successful in treating a recurrent tumor in one patient’s eye. Additionally, Yi et al. [6] recently reported one case of globe salvage in CBME treated with combined intra-arterial chemotherapy and cryotherapy. Cryotherapy is a well-established treatment modality in retinoblastoma, proven to eradicate up to 90% of treated tumors [16]. In this technique, performed under indirect ophthalmoscopic guidance, a cryoprobe is applied directly to the ocular surface adjoining the tumor. Three cycles of freezing and thawing are applied, with care to encompass the whole tumor mass within the forming iceball. Typical complications of cryotherapy include ocular tenderness, transient conjunctival edema, and vitreous hemorrhage, but retinal tears and serous retinal detachment have also been reported [17,18]. The options for retinal detachment surgery in children include scleral buckling and vitrectomy, with a preference for the former because of the unique anatomy of the pediatric eye and the lower risk of cataract-induced amblyopia [19].

In the course of treatment, retinal detachment occurred in our patient, and the complication was successfully managed with scleral buckling. The tumor was effectively controlled with cryotherapy and has remained in regression during 6 months of follow-up since the last treatment.

The prognosis in CBME without extrabulbar extension is favorable. Broughton and Zimmermann [2] reported four tumor-related deaths in a cohort of 56 patients with CBME (follow-up was available for 33 patients). All four of these patients had had evidence of orbital involvement. Kaliki et al. [1] reported regional lymph node metastasis in 8% of patients, all of whom showed extraocular tumor extension without tumor-related deaths. Canning et al. [5] reported no deaths, and 100% of the patients remained recurrence-free during a follow-up period of up to 11 years. During more than 5 years under our care, our patient remained under systematic oncological control with regular imaging tests, and no CBME or PPB recurrences or metastases were discovered.

## 4. Conclusions

Pediatric ophthalmologists and oncologists must be aware of the risk of CBME in patients with *DICER1* mutation and of the possibility of bilateral disease. Scheduling regular ophthalmological visits for the affected individuals could increase the chance of detection of new tumors at an early stage when globe-sparing methods of treatment can be effective. Utilizing several imaging modalities can help assess the disease progression with higher accuracy, facilitating decision making and enhancing the likelihood of tumor control.

## Figures and Tables

**Figure 1 diagnostics-15-00694-f001:**
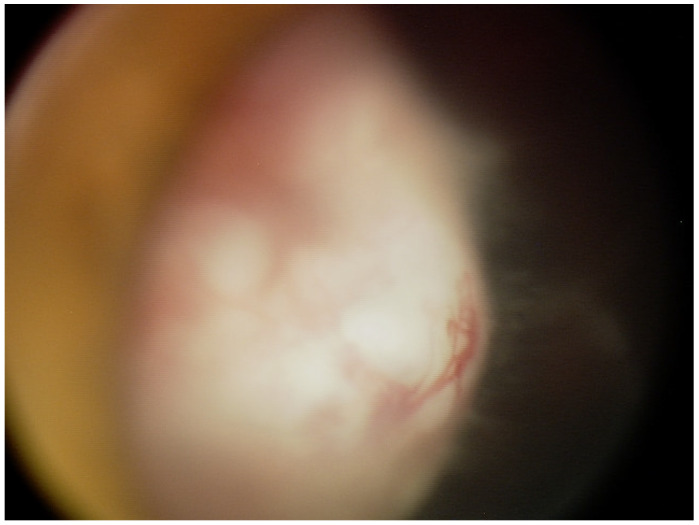
A retrolental whitish-pink mass with visible blood vessels and vitreous seeding.

**Figure 2 diagnostics-15-00694-f002:**
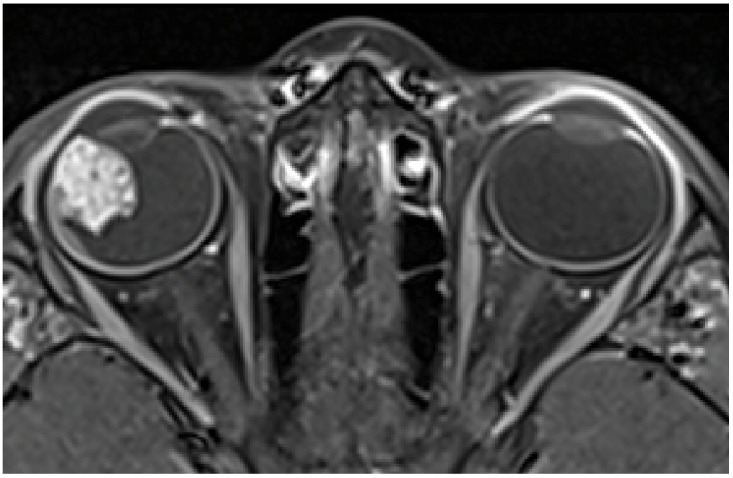
MRI showing a ciliary body mass with heterogenic intensity in the right eye.

**Figure 3 diagnostics-15-00694-f003:**
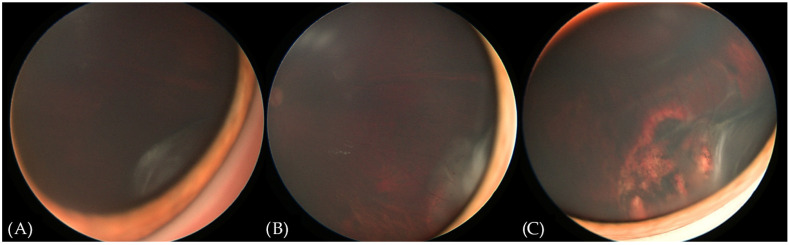
RetCam (Envision) fundus photographs showing the evolution of the lesion in the left eye. (**A**) New flat whitish lesion in the infero-temporal ora serrata. (**B**) Enlargement of the lesion and the occurrence of abnormal blood vessels. (**C**) Appearance of the lesion upon the last follow-up. See the fibrotic changes and the chorioretinal scar anterior to the edge of the scleral buckle.

## Data Availability

This article does not include any additional primary data besides the information already presented in the case report section.

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
