# Peer review of "Presumed Bilateral Ciliary Body Medulloepithelioma in a Child with Pleuropulmonary Blastoma and DICER1 Mutation"

_diagnostics, 2025, doi:10.3390/diagnostics15060694_

Round 1
Reviewer 1 Report
Comments and Suggestions for Authors
Dear authors, please find my review comments in the document attached. The comment, will help improve the quality of your manuscript.

There are errors in the English language.
Please employ the services of an English language editor to improve the quality of your manuscript.
Reviewer 2 Report
Comments and Suggestions for Authors
In the current manuscript, authors have submitted a case report involving a 6-year-old boy diagnosed with CBME with a previous history of pleuropulmonary blastoma type II. In addition, the subject has been confirmed to carry DICER1 mutation. The reviewer has found the case report as a first of its kind with a rare case of having presumed bilateral CBME with DICER1 mutation coexisting with pleuropulmonary blastoma. The reviewer has following suggestions to improve the submitted manuscript.
Please detail the sequence of interventions or timeline performed including imaging in the results section.
Also, the treatment procedures can be elaborated more justifying the reason for opting transscleral cryotherapy and subsequent scleral buckling treatment procedure.
Authors can expand more on similarities on the clinical manifestations of the disease with previous reports.
The case report included 6-month follow-up strategy by ophthalmological exams. Please elaborate more on possible long-term implications or success in such cases.
Major limiting factor of the report is detailed discussion on the mechanisms relating DICER1 mutation and the pathogenesis of CBME is missing. Contribution of DICER1 as a prominent genetic risk factor should be discussed further to dissect its function in the future research.
